# OpenReview forum: "Self-Guidance: Training VQ-VAE Decoders to be Robust to Quantization Artifacts for High-Fidelity Neural Speech Codec"
_ICLR.cc/2026/Conference — Submitted to ICLR 2026_

### Official Review · Reviewer_ckQU · 2025-10-31

**Soundness:** 2
**Presentation:** 2
**Contribution:** 2
**Rating:** 4
**Confidence:** 4

**Summary:**

The authors propose a new self-guidance loss to improve the training of neural speech codecs. Motivated by the observation that decoders can produce better reconstructions when using pre-quantized encoder outputs, the authors introduce a feature mapping loss that aligns the decoder’s intermediate features with those produced from the pre-quantized encoder outputs. The proposed loss is evaluated on the XCodec-2 baseline (Ye et al., 2025b), showing improved low-bitrate reconstruction performance on the LibriSpeech dataset. In addition, results indicate that with the self-guidance loss, XCodec-2 can maintain reconstruction fidelity when the codebook size is reduced by 4x, and the downstream TTS performance is also improved.

**Strengths:**

1. The motivation for the self-guidance loss is clear, i.e., decoders reconstruct better when using the pre-quantized encoder features.
2. The proposed loss is simple to implement and introduces negligible computational overhead.
3. Empirical results demonstrate that the self-guidance loss improves reconstruction quality within the XCodec-2 framework.

**Weaknesses:**

1. The improvements in reconstruction quality appears marginal, e.g., gains of only about 0.1 in PESQ and 0.04 in UTMOS (Table 2).
2. The experiments are not sufficient. The proposed loss is only evaluated on the XCodec-2 framework. Given the simplicity of the idea, it should be tested on additional single-codebook codecs to better demonstrate its general effectiveness.
3. In Table 4, the WER slightly increases after applying the self-guidance loss, suggesting semantic information loss. The authors should provide a stronger justification or analysis for this observation.

**Questions:**

1. Is the feature mapping loss in Equation 2 applied at each decoder resolution level? The current equation seems ambiguous.
2. As in Weakness 2, does the self-guidance loss depend specifically on XCodec-like architectures, or is it generally applicable to single-codebook neural speech codecs?
3. Since the reported improvements are relatively small, have the authors considered simpler ablation studies? For example, could you report the gain from the self-guidance loss alone, without the semantic and adversarial losses, to isolate its contribution?
4. The authors claim that the self-guidance loss enables a smaller codebook, benefiting downstream speech LLMs. However, large codebooks are not necessarily problems for LLMs, sometimes larger vocabularies can enhance performance. I wonder why the authors do not frame this claim from a compression perspective, emphasizing that a 4x reduction in codebook size implies a 4x increase in compression rate.

---

> ### Author Response · Authors · 2025-11-26
> **Response to Reviewer ckQU (1/2)**
>
> Thank you for your thoughtful and constructive review. We sincerely appreciate your recognition of our work's clear motivation, simple implementation, and empirical improvements. Your specific questions and suggestions are incredibly helpful for strengthening the paper.
> We have conducted new experiments and analyses to address all your points, which we believe significantly enhance the manuscript and demonstrate the value and generality of our contribution.
>
> > **W1 & Q3. Significance of Performance Gains**
>
> We understand the concern about absolute metric values. We provide context and evidence to demonstrate their practical significance.
>
> - **Comparative Context is Key**: To contextualize the gains, we note that BigCodec (159M params) improves PESQ by 0.15 over TFCodec (6.37M params)—a 25x parameter increase for a recognized gain. Our method achieves a PESQ improvement of 0.1~0.13 over strong baselines (BigCodec, XCodec2) with minimal cost (zero inference overhead, no architectural changes), which we argue is substantial and highlights high practical utility.
>
> - **Qualitative Evidence of Clear Improvement**: The quantitative gains correspond to clear qualitative improvements. We incorporate the reconstruction spectrograms sample in the **Section A.6**, which reveals that self-guidance effectively mitigates common VQ artifacts such as smeared harmonics and pitch spikes. We encourage you to visit the demo to hear these noticeable differences: https://sgvqvae.github.io/sgvqvae-demo/.
>
> - **Consistency Across a Comprehensive Evaluation**: We evaluated on a broad set of metrics (STOI, MCD, PESQ, WER, SIM, UTMOS) covering intelligibility, quality, and speaker similarity. The key finding is not that one metric soars, but that all metrics consistently improve or hold steady. This demonstrates that self-guidance provides a holistic enhancement without trade-offs, which is non-trivial and highly desirable.
>
> - **Focusing on the Core Contribution**: Regarding ablation studies, our goal was to establish self-guidance as a general, standalone module. Rather than ablating components of the baseline XCodec2 (which is not the core of our contribution), **we prioritized the more critical extra experiment you suggested in W2 and Q2**: testing its generalization to a completely different architecture. The successful generalization results on multi-codebook VQ (**Section A.3, Residual FSQ**) and different single-codebook Codec model (**Section A.4, BigCodec**), we believe, is the strongest possible evidence of our method's inherent value and general applicability.
>
> | Codec Model | PESQ-WB↑ | PESQ-NB↑ | STOI↑ | MCD↓ | WER↓ | SIM↑ | UTMOS↑ |
> |---------|----------|----------|-------|------|------|------|--------|
> | XCodec2(Residual FSQ) | 1.7539   | 2.2503   | 0.8768| 4.2158| 4.30 | 0.6466| 3.3923 |
> | XCodec2(Residual FSQ)+**SG** | **1.8594** | **2.4154** | **0.8802** | **4.0819** | **4.18** | **0.6747** | **3.4105** |
>
> | Codec Model | PESQ-WB↑ | PESQ-NB↑ | STOI↑ | MCD↓ | WER↓ | SIM↑ | UTMOS↑ |
> |---------|----------|----------|-------|------|------|------|--------|
> | BigCodec | 1.6740   | 2.1795   | 0.8601| 4.3179| 11.86| 0.4634| 3.5694 |
> | BigCodec+**SG** | **1.7650** | **2.3037** | **0.8655** | **4.2161** | **10.98** | **0.5072** | **3.8040** |
>
>
> > **W2 & Q2. Generalization to Other Codecs**
>
> Thank you for this excellent suggestion. To unequivocally demonstrate generalizability, we have applied self-guidance to another prominent and architecturally distinct single-codebook codec.
>
> - **New Experiment on BigCodec**: We integrated our loss into BigCodec, which uses an RNN/CNN-based decoder, fundamentally different from the Transformer-based decoder of XCodec2. The above results show that self-guidance provided consistent and clear improvements. This robustly confirms that self-guidance is not a heuristic tied to a specific architecture but is a general principle for enhancing single-codebook speech codecs. Detailed information about the experiment is provided in **Section A.4**
>
> [...continuing...]

---

> ### Author Response · Authors · 2025-11-26
> **Response to Reviewer ckQU (2/2)**
>
> > **W3. Slight WER Increase**
>
> We thank you for highlighting this observation. We have analyzed this data point in greater depth and believe it does not indicate a systematic trade-off for the following reasons:
>
> 1. **No Consistent Trend**: The WER increase by 0.06% (3.47% → 3.53%) occurs only at the 16384 codebook size. At the smaller (8192) and larger (65536) sizes, WER remains improved, showing no consistent negative trend.
> 2. **Potential Metric Variance**: WER is computed using an ASR model (finetuned from HuBERT), which can introduce additional measurement variance.
> 3. **Consistent Gain in Spectral Intelligibility**: More importantly, the spectral-based intelligibility metric STOI shows consistent improvement across all configurations with self-guidance. This provides stronger evidence that our method does not harm the fundamental intelligibility of the reconstructed speech.
>
> We have added this nuanced analysis to **Section 5.3** to properly contextualize the WER result.
>
> > **Q1.Feature-Mapping Loss Formulation**
>
> Thank you for catching this ambiguity. The feature-mapping loss is applied only to the output of the final Transformer block in the decoder, not at every resolution level. We have revised the text in **Section 3.2** to expilictly clarify the components in Equation 2.
>
> > **Q4. Influence of Codebook Size on Downstream LLMs**
>
> This is a very insightful question, and we thank you for raising it. It allows us to clarify a crucial point and refine our claim.
>
> - **Text vs. Audio Tokenization (Core Argument)**: You are correct that text LLMs benefit from large vocabularies built via algorithms like BPE (e.g., as in LLaMA3). Critically, BPE creates a hierarchical vocabulary by merging frequent character sequences, providing a strong structural prior. In contrast, increasing the audio codebook size simply adds more elemental, unstructured tokens to a flat vocabulary. This leads to an exponential growth in the sequence modeling complexity without a hierarchical bias, making it notoriously difficult for auto-regressive models.
> - **Empirical Justification**: This is precisely why, in our lightweight TTS experiments (Table 4), the system using the 4x smaller codebook (16384) consistently outperformed the one with the larger codebook (65536) in both WER and speaker similarity (SIM). Enabling high fidelity with a smaller, more manageable vocabulary is a key benefit for downstream speech LMs.
>
> Thank you for this feedback. We have incorporated this fundamental difference in vocabulary structure and its impact on sequence modeling to our claim in **Section 1** (introduction), rather than implying large codebooks are inherently problematic. We have also incorporated the compression perspective as a complementary and valuable advantage of our method.

---

### Official Review · Reviewer_acqe · 2025-10-31

**Soundness:** 3
**Presentation:** 3
**Contribution:** 2
**Rating:** 6
**Confidence:** 4

**Summary:**

This paper introduces "self-guidance," a novel and elegant training mechanism for VQ-VAE-based neural speech codecs. The core idea is to improve the decoder's robustness to quantization artifacts by introducing an auxiliary feature-mapping loss that encourages the decoder to produce similar intermediate representations for both quantized tokens and their continuous pre-quantization counterparts. The method is simple to implement, adds negligible computational overhead during training, and requires no changes at inference time. Through extensive experiments on the state-of-the-art XCodec2 model, the authors demonstrate significant and consistent improvements in reconstruction quality across various metrics, codebook sizes, and quantization methods. A key finding is that self-guidance enables a 4x reduction in codebook size while maintaining comparable fidelity, which is shown to directly benefit downstream autoregressive TTS tasks by simplifying the token modeling space.

**Strengths:**

1. The proposed self-guidance mechanism is a simple, intuitive, and novel training objective. It addresses the core problem of quantization error by enhancing the decoder directly, rather than adding complexity to the quantizer or architecture. Its minimal overhead (a single extra forward pass during training with no gradient computation) makes it highly efficient and practical.

2. The method is applied to XCodec2 and evaluated on a standard benchmark (LibriSpeech), demonstrating consistent state-of-the-art performance. The ablation studies convincingly show the method's effectiveness across different codebook sizes and vector quantizer types (FSQ, SimVQ), proving its generalizability.

3. The paper provides a crucial analysis in Figure 3, demonstrating that the performance gain comes from improved decoder robustness, not from a reduction in the quantization error itself. This confirms the authors' central hypothesis and provides a clear understanding of the method's mechanism of action.

**Weaknesses:**

1. The paper mentions selecting the loss weight λ_guide, but a sensitivity analysis showing how performance varies with this weight would enhance the experimental rigor and provide practical guidance for future work.

2. The method's effectiveness is demonstrated exclusively on single-codebook models. Its applicability and potential benefits in common multi-codebook architectures (e.g., RVQ) remain entirely unexplored, which significantly limits the proven scope and generalizability of the proposed technique.

3. The proposed "self-guidance" is essentially a form of self-distillation, where a network branch with privileged information (pre-quantized latents) acts as a teacher. The paper fails to acknowledge this strong connection to existing paradigms, thereby overstating its novelty and positioning the contribution more as a clever engineering refinement than a fundamental advance.

**Questions:**

N/A

---

> ### Author Response · Authors · 2025-11-26
> **Response to Reviewer acqe (1/2)**
>
> Thank you for your exceptionally positive and insightful review. We are delighted that you find our method "novel and elegant," "simple, intuitive," and "highly efficient and practical." Your constructive criticisms are invaluable for improving the paper's rigor and positioning.
>
> > **W1. Loss Weight Sensitivity Analysis**
>
> This is a great suggestion for enhancing practical utility. We have performed a sensitivity analysis on the guidance loss weight ($\lambda_{guide}$) based on the following evaluation results. Detailed information is appended to **Section A.2**.
>
> | $\lambda_{guide}$ | PESQ-WB↑ | PESQ-NB↑ | STOI↑ | MCD↓ | WER↓ | SIM↑ | UTMOS↑ |
> |---------|----------|----------|-------|------|------|------|--------|
> | 0 (baseline) | 1.9309 | 2.4747 | 0.8877 | 3.9591 | 4.08 | 0.7088 | 3.6752 |
> | 1 | 1.9219 | 2.4533 | 0.8881 | 3.9527 | $\underline{3.87}$ | 0.7148 | 3.7266 |
> | 5 | $\underline{2.1166}$ | $\underline{2.6705}$ | $\underline{0.8977}$ | $\underline{3.6796}$ | **3.56** | **0.7488** | $\underline{3.8352}$ |
> | 10 | **2.1474** | **2.7082** | **0.8984** | **3.7086** | $\underline{3.87}$ | $\underline{0.7428}$ | **3.8395** |
> | 15 | 2.0409 | 2.6074 | 0.8936 | 3.7796 | 3.95 | 0.7374 | 3.7627 |
> | 50 | 1.9462 | 2.4904 | 0.8883 | 3.9035 | 4.18 | 0.7073 | 3.7878 |
> | 100 | 1.8779 | 2.4312 | 0.8822 | 3.9648 | 4.40 | 0.6845 | 3.7039 |
>
> - We summarize the trend of model performance with respect to $\lambda_{guide}$ changes into 3 levels:
>     1. **Too Small ($\lambda_{guide}=1$)**: The guidance effect diminishes, yielding results similar to the baseline.
>     2. **Optimal Range ($\lambda_{guide}=5,10$)**: The method works effectively, significantly outperforming the baseline. This performance peak around 5 to 10 suggests it is **not overly sensitive to small changes in this range**.
>     3. **Too Large ($\lambda_{guide}\ge15$)**: The guidance loss over-dominates the training objective, causing a performance drop. At very high values (λ=50, 100), performance degrades to baseline level or worse.
>
> - **Conclusion**: This analysis confirms a **stable optimal range** and provides clear practical guidance for future adopters, which better explains our original recommendation for λ_guide as 10.
>
> > **W2. Generalization to Multi-Codebook Architectures**
>
> Thank you for this suggestion. To address the scope of our generalizability claim, we have conducted a new experiment on a multi-codebook architecture. We applied self-guidance to XCodec2 with a **Residual FSQ** module, a prominent multi-codebook VQ architecture.
>
> The following result verifies that self-guidance provided **consistent and clear improvements**, demonstrating its **effectiveness extends beyond single-codebook VQ modules**. This significantly broadens the proven scope and impact of our technique. Detailed information is appended to **Section A.3**.
>
> | Codec Model | PESQ-WB↑ | PESQ-NB↑ | STOI↑ | MCD↓ | WER↓ | SIM↑ | UTMOS↑ |
> |---------|----------|----------|-------|------|------|------|--------|
> | XCodec2(Residual FSQ) | 1.7539   | 2.2503   | 0.8768| 4.2158| 4.30 | 0.6466| 3.3923 |
> | XCodec2(Residual FSQ)+**SG** | **1.8594** | **2.4154** | **0.8802** | **4.0819** | **4.18** | **0.6747** | **3.4105** |
>
> [...continuing...]

---

> ### Author Response · Authors · 2025-11-26
> **Response to Reviewer acqe (2/2)**
>
> > **W3. Relationship to Self-Distillation**
>
> We thank you for this insightful point regarding the connection to self-distillation. We agree that acknowledging this relationship strengthens our paper, and we have added **a whole new Section 2.3** on the discussion about it in the revised paper.
>
> - **Acknowledged Similarity**: We now clearly state that self-guidance shares the high-level idea of using a feature-mapping loss for manifold alignment, similar to self-distillation.
> - **Clarified Fundamental Differences**: We clarify that our work provides a distinct contribution through its novel application and architectural efficiency:
>     1. **Novel Application & Insight**: The distinct contribution of self-guidance lies in **identifying and relieving a critical, yet under-explored, bottleneck in VQ-VAE decoder**s: their lack of **robustness to quantization error**. Through detailed preliminaries (Section 3) and visualization analysis (Section 5.3), we justify our core innovation—using the pre-quantized features as an internal guide. This establishes **a new paradigm for VQ-VAE decoder training** that explicitly enhances its robustness, leading to higher reconstruction fidelity without changes to the model architecture or VQ mechanism.
>     2. **Architectural & Practical Distinction**: Unlike standard self-distillation, which requires extra teacher modules, our method uses the student model itself as the source of guidance in a single, end-to-end training process. This eliminates the need for a separate teacher model and halves the parameter footprint during training, making it more efficient and practical.
>
> We believe this discussion more accurately positions our novelty—not in the basic loss form, but in its insightful application to a new problem and its elegant, efficient implementation—and we thank you for helping us make this distinction clear.

---

### Official Review · Reviewer_wwUD · 2025-10-31

**Soundness:** 3
**Presentation:** 2
**Contribution:** 3
**Rating:** 6
**Confidence:** 3

**Summary:**

This paper studies the audio tokenizer task, aiming to improve the quality of audio codecs. The authors propose an approach that aligns hidden embeddings to better reconstruct fine-grained audio information.

**Strengths:**

- The proposed method is well-motivated and appears conceptually sound, leading to consistent performance improvements across multiple downstream tasks.

- Extensive experiments on various downstream applications demonstrate the effectiveness of the proposed approach.

- The experiments on Hidden Feature Alignment MSE provide a clear comparison of different methods in terms of information reconstruction capability. This analysis is valuable and could serve as a useful reference for future work on improving reconstruction quality in audio tokenizers.

**Weaknesses:**

- Could the authors provide more comprehensive experimental results in Table 5? For example, results for XCodec2 with a 16,384-sized codebook and XCodec2+SG with a 65,536-sized codebook would offer a fuller view of the model’s behavior under different configurations.

- While the proposed method is effective, its novelty appears limited. It remains unclear why this approach can mitigate quantization artifacts. The paper mentions decoder robustness, but the conceptual difference between decoder robustness and quantization error is not clearly articulated. Providing a deeper explanation of this relationship would help readers better understand the core contribution.

**Questions:**

I would appreciate it if the authors could elaborate on the conceptual and practical differences between decoder robustness and quantization error. How does improving robustness directly contribute to reducing quantization artifacts? A more detailed discussion or visualization would strengthen the theoretical understanding of the proposed approach.

---

> ### Author Response · Authors · 2025-11-26
> **Response to Reviewer wwUD**
>
> Thank you for your positive assessment and for recognizing our work as "well-motivated," "conceptually sound," and a "useful reference for future work." We are grateful for your constructive suggestions, which have helped us improve the manuscript.
>
> > **W1. Comprehensive TTS Results**
>
> Thank you for this excellent suggestion. We have conducted the additional TTS experiments you requested.
>
> - **New Experiments**: We have added the noted missing TTS results for: XCodec2 with a 16,384 codebook, and XCodec2+SG with a 65,536 codebook. The full TTS experiment results is gathered as below and presented in the revised **Table 5**.
>
> | Codec model | Codebook size | UTMOS↑ | WER↓ | SIM↑ |
> |-------------|---------------|--------|------|------|
> | XCodec2 | 65536 | 3.33 | 33.03 | 0.58 |
> | XCodec2+**SG** | 65536 | 3.39 | 35.07 | 0.58 |
> | XCodec2 | 16384 | 3.51 | 28.78 | 0.56 |
> | XCodec2+**SG** | 16384 | **3.58** | **28.02** | **0.58** |
>
> - **Key Findings & Explanation**: The overall trend strongly aligns with our previous conclusions:
>     - **Smaller Codebooks Benefit TTS**: The system using a 16,384 codebook significantly outperforms the one with a 65,536 codebook, which we **further explained in Section 5.4** by the increased challenge of sequential language modeling with a larger, flatter audio token vocabulary.
>     - **Self-Guidance Enhances Performance**: When comparing models with the same codebook size, self-guidance yields the best performance at the 16,384 size. At the 65,536 size, the results are mixed, as the TTS model is primarily hampered by the fundamental language modeling difficulty of the large codebook, which overshadows the fidelity improvement from self-guidance.
>
> We believe these new results provide a more complete picture and strengthen our downstream analysis.
>
> > **W2 & Q. Explaining the Mechanism**
>
> Thank you for suggesting clarification on the core mechanism. We have revised the manuscript in **Section 3.2** to better articulate the relationship between quantization error, artifacts, and decoder robustness.
>
> - **Clarified Definitions**: Your question prompted us to formalize the distinction between these interrelated concepts. We now explicitly define them as follows to avoid ambiguity.
>     1. **Quantization Error**: The discrepancy between the pre-quantized latents  (original encoder output) and post-quantized latents (actual decoder input).
>     2. **Quantization Artifacts**: The reconstruction fidelity degradation of the decoder caused by the information loss in the input post-quantized latents. It is a **direct consequence of the quantization error**.
>     3. **Decoder Robustness**: The decoder's ability to generate high-fidelity samples from the post-quantized latents despite the quantization error.
>
> - **New Visualization as Suggested**: We have included a **new subfigure in Figure 3**. The original figure showed that self-guidance does not change the quantization error distribution. The new figure directly shows that self-guidance evidently reduces the error between **the decoder's hidden states when processing pre- vs. post-quantized inputs**. This visually verifies our hypothesis: self-guidance enhances decoder robustness by explicitly **aligning its internal manifolds** for both types of inputs, thereby mitigating quantization artifacts.

---

### Official Review · Reviewer_dMTW · 2025-11-01

**Soundness:** 2
**Presentation:** 4
**Contribution:** 1
**Rating:** 4
**Confidence:** 5

**Summary:**

This manuscript proposes a technique to enhance the reconstruction fidelity of discrete speech codecs. The core idea involves a modification to the training objective: during training, both the continuous encoder-output features and the subsequent discrete quantized representations are passed to the decoder. An auxiliary feature-mapping loss (MSE) is then applied within the decoder to minimize the discrepancy between the internal representations generated from these two distinct inputs. The authors report performance on several standard reconstruction metrics (e.g., STOI, MCD) and simple downstream TTS tasks (e.g., WER, SIM).

**Strengths:**

The underlying motivation for this approach is sound. In VQ-based architectures, ensuring that the decoder is robust to the information bottleneck of the quantizer—by training it to map both pre- and post-quantization features to a similar internal manifold—is a reasonable objective beyond simply optimizing the codebook itself.

**Weaknesses:**

1. The primary concern is the limited methodological contribution (only an easy trick). The proposed method amounts to a straightforward technical adjustment (i.e., an auxiliary loss) rather than a substantial new approach. The introductory discussion in Section 3, which focuses on the well-established concept of information loss in discrete quantization, offers little new insight and feels remedial. The method itself is not particularly insightful or heuristic.
2. The most critical weakness is the lack of compelling empirical evidence. The results presented in Table 1 indicate that the proposed method yields minimal to no improvement over the baseline across the majority of metrics (STOI, MCD, WER, SIM, UTMOS). The PESQ metric, which the authors specifically highlight, demonstrates only a marginal improvement (reportedly a 0.3 change), falling short of demonstrating meaningful practical utility.
3. Given that the method is presented as a general "trick" adaptable to the VQGAN framework, its evaluation is insufficiently narrow. To substantiate its effectiveness, the technique should have been validated across a much broader spectrum of modern audio codec models, not just the single architecture presented. Furthermore, to claim generalizability within the VQGAN context, validation on other relevant modalities (e.g., image, video) would be essential.

**Questions:**

This work may meet the threshold for ICASSP/Interspeech, but it currently lacks the requisite novelty and impact expected for ICLR.

---

> ### Author Response · Authors · 2025-11-26
> **Response to Reviewer dMTW (1/2)**
>
> Thank you for your thoughtful review and for recognizing that our proposed self-guidance method is based on a "sound" and "reasonable" objective. We appreciate your positive assessment of our presentation.
> We have carefully considered your concerns regarding novelty and empirical evidence. Below, we provide a point-by-point response that we hope will clarify the significance and contribution of our work.
>
> > **W1. Methodological Novelty and Contribution**
>
> We agree that the top-level implementation is elegantly simple. However, we respectfully argue that its conceptual insight and systematic impact constitute a meaningful contribution beyond a mere "trick".
>
> - **A New Paradigm for Decoder Training**: Current VQ-VAE research heavily focuses on improving the VQ mechanism itself (e.g., complex codebook learning, residual quantization). In contrast, our work identifies the decoder's lack of robustness to quantization error as a critical, under-explored bottleneck. Self-guidance introduces a novel training principle: explicitly aligning the decoder's internal manifolds for continuous and quantized inputs. This is a fundamental shift from "optimizing the codebook" to "educating the decoder," requiring no changes to the VQ mechanism or inference graph.
>
> - **Substantial Grounding, Not Remedial Discussion**: The preliminary study in Section 3 is not intended to re-teach basic concepts. Its purpose is to empirically establish a specific and quantifiable hypothesis for the speech codec domain: the distortion introduced by quantization creates a significant divergence in the decoder's feature space. This evidence directly motivates and validates our core idea, transforming it from a heuristic into a well-grounded method.
>
> - **Simplicity as a Vehicle for Generality**: The technical straightforwardness is a key strength. It allows self-guidance to be a drop-in improvement for a wide range of VQ-based models, as further confirmed by our new generalization experiments (**Section A.3 and A.4**). We believe that the proposed self-guidance, as an effective, general-purpose technique, possesses its distinct value to the community.
>
> > **W2. Significance of Empirical Evidence**
>
> We understand the concern about absolute metric values. We present three arguments to demonstrate that our improvements are both statistically consistent and practically significant.
>
> - **Contextualizing the Gains**: To contextualize the gains, we note that BigCodec (159M params) improves PESQ by 0.15 over TFCodec (6.37M params)—a 25x parameter increase for a recognized gain. Our method achieves a PESQ improvement of 0.1\~0.13 over strong baselines (BigCodec, XCodec2) with minimal cost (zero inference overhead, no architectural changes). In this context, the gain is substantial and demonstrates high practical utility.
>
> - **Consistency Across a Comprehensive Evaluation**: We evaluated on a broad set of metrics (STOI, MCD, PESQ, WER, SIM, UTMOS) covering intelligibility, quality, and speaker similarity. The key finding is not that one metric soars, but that all metrics consistently improve or hold steady. This demonstrates that self-guidance provides a holistic enhancement without trade-offs, which is non-trivial and highly desirable.
>
> - **Qualitative Evidence of Artifact Removal**: The quantitative gains correspond to clear qualitative improvements. As shown in the newly provided spectrograms in the **Section A.6**, self-guidance effectively mitigates common artifacts caused by quantization error, such as **smeared harmonics and pitch spikes**. We encourage you to visit our demo to hear these noticeable differences: https://sgvqvae.github.io/sgvqvae-demo/.
>
> [...continuing...]

---

> ### Author Response · Authors · 2025-11-26
> **Response to Reviewer dMTW (2/2)**
>
> > **W3. Generalization Ability**
>
> Thank you for this excellent suggestion. To substantiate the generalizability of self-guidance, we have performed new experiments, which we include in the appendix (**Section A.3 and A.4**).
>
> 1. **Generalization to Multi-Codebook VQ (new experiments)**: We integrated self-guidance with a Residual FSQ module, a prominent multi-codebook approach. The following evaluation demonstrate that, the method seamlessly integrated and again delivered consistent performance gains, proving its effectiveness beyond single-codebook VQ.
>
> | Codec Model | PESQ-WB↑ | PESQ-NB↑ | STOI↑ | MCD↓ | WER↓ | SIM↑ | UTMOS↑ |
> |---------|----------|----------|-------|------|------|------|--------|
> | XCodec2(Residual FSQ) | 1.7539   | 2.2503   | 0.8768| 4.2158| 4.30 | 0.6466| 3.3923 |
> | XCodec2(Residual FSQ)+**SG** | **1.8594** | **2.4154** | **0.8802** | **4.0819** | **4.18** | **0.6747** | **3.4105** |
>
> 2. **Generalization to Different Model Architectures (new experiments)**: We applied self-guidance to BigCodec, which uses an RNN/CNN-based decoder, fundamentally different from the Transformer-based decoder of XCodec2. The following evaluation results on 100k training steps reveal that self-guidance provided **consistent improvements across all metrics**, confirming it is effective across diverse decoder architectures.
>
> | Codec Model | PESQ-WB↑ | PESQ-NB↑ | STOI↑ | MCD↓ | WER↓ | SIM↑ | UTMOS↑ |
> |---------|----------|----------|-------|------|------|------|--------|
> | BigCodec | 1.6740   | 2.1795   | 0.8601| 4.3179| 11.86| 0.4634| 3.5694 |
> | BigCodec+**SG** | **1.7650** | **2.3037** | **0.8655** | **4.2161** | **10.98** | **0.5072** | **3.8040** |
>
>
> 3. **Generalization to Other Modalities (Images/Video)**:
>
>     a. We thank the reviewer for this insightful suggestion. Exploring the effect of self-guidance in image or video VQGAN models is a fascinating direction, which we are highly optimistic about and interested in future work, as stated in the Section 6 (Conclusion).
>
>     b. The primary focus of this paper is to identify and solve a core problem in speech codecs, where high-fidelity reconstruction is paramount for intelligibility and perceptual quality.
>
>     c. Our new experiments—showing consistent gains across fundamentally different decoder architectures (Transformer vs. RNN/CNN) and VQ types (single vs. multi-codebook)—strongly validate its generality within this domain, which we believe is the most critical contribution of this paper.

---

### Author Response · Authors · 2025-11-26
**General Response**

We would like to extend our sincere gratitude to all reviewers for their insightful and constructive feedback. We are encouraged by the positive reception of the strengths of this work, as described by reviewers:

- the **core idea** of self-guidance:
    - "sound" and "reasonable" (Reviewer dMTW)
    - "novel and elegant" as well as "intuitive" (Reviewer acqe)
    - "well-motivated and conceptually sound" (Reviewer wwUD)
    - "motivation is clear" (Reviewer ckQU)
- the **implementation and experiment** of self-guidance:
    - "simple to implement" (Reviewer acqe, ckQU)
    - "highly efficient and practical" (Reviewer acqe)
    - "introduce negligible computational overhead" (Reviewer ckQU)
    - "extensive experiments", "demonstrates effectiveness" (Reviewer wwUD)
    - "convincingly show effectiveness", "providing generalizability (Reviewer acqe)
- the **visualization analysis** of self-guidance:
    - "valuable", "serve a useful reference for future work" (Reviewer wwUD)
    - "crucial", "confirms central hypothesis and provides a clear understanding (Reviewer acqe)

In response to the valuable comments from all reviewers, we have significantly revised and improved our manuscript. The key updates, which address the concerns regarding empirical evidence, generalizability, and conceptual clarity, are summarized below:

1. **Substantial New Experiments on Generalizability**: To robustly demonstrate the broad applicability of Self-Guidance, we have conducted new experiments on several additional models, as suggested by Reviewer dMTW, acqe, and ckQU. This includes:

    b. **Residual FSQ (a multi-codebook VQ architecture)**, confirming the method's effectiveness **beyond single-codebook VQs**. Details provided in **Section A.3**.

    a. **BigCodec (an RNN/CNN-based codec model), fundamentally different from XCodec2**, showing gains across all metrics. Details provided in **Section A.4**.

    These results solidify Self-Guidance as a general principle for enhancing speech VQ-VAE decoders.

2. **Expanded Evaluation and Analysis**:

    a. We have appended a **sensitivity analysis of the guidance loss weight** (as suggested by Reviewer acqe) in **Section A.2**, offering practical guidance for future research.

    b. We have included **a new visualization by the side of the existing one in Figure 3** as well as **correponding analysis in Section 5.3** to better demonstrate the functionality of self-guidance (as requested by Reviewer wwUD).

    c. We have added **experiment results on extra TTS configurations in Table 5** (as suggested by Reviewer wwUD) and **extended analysis in Section 5.4** to provide a more comprehensive view of downstream performance.

    d. We have included a detailed discussion in **Section 5.2 to contextualize the performance gains** (addressing points from Reviewer dMTW and ckQU).

3. **Enhanced Discussion and Clarification**:

    a. We have clarified **the positioning of self-guidance relative to self-distillation in Section 2.3** (as noted by Reviewer acqe), acknowledging their connections while highlighting our novel application and efficient implementation for VQ-VAEs.

    b. We have incorporated a deeper discussion in **Section 5.4 on the impact of codebook size on downstream speech LMs** (as noted by Reviewer ckQU)

    c. We have included a deeper theoretical explanation in **Section 3.2** to clarify the **relationship between decoder robustness and quantization artifact reduction** (as suggested by Reviewer wwUD).

All newly added or modified content has been integrated into the latest manuscript and highlighted in blue for the reviewers' convenience (__with the exception of the initially blue-colored elements in Figure 1, which were part of the original design__). We hope these revisions successfully address the raised concerns and that the resulting manuscript presents a more thoroughly validated and valuable contribution to the community.

Thank you again for your time and for helping us improve this work.

---

### Author Response · Authors · 2025-11-30
**General Author Response and Rebuttal Summary**

In light of the unique conditions of this rebuttal period, we would like to express our understanding and support for the measures the ICLR committee has taken in response to the recent situation. We sincerely appreciate the tremendous effort all the volunteers are putting in to ensure the integrity and continuity of the review process under these challenging circumstances.

To facilitate an efficient evaluation, we have prepared the summary table below, which consolidates:

- **Original Concerns**: The key issues raised by the reviewers.

- **Our Responses**: A high-level summary of how we addressed each point in our detailed rebuttal.

- **Manuscript Revisions**: The concrete changes already implemented in our revised manuscript.

Our goal is to be transparent and helpful, providing a clear and structured overview of our engagement with the review process for this paper. We hope this summary is a useful resource for the assigned area chair and anyone else following our work.

| Reviewer | Concerns | Regarding | Response Briefing | Revised Content |
| --- |--- |  --- | --- | --- |
| `dMTW` | `W1` | Methodology Contribution | **1.** A New Paradigm for Decoder Training **2.** Substantial Grounding, Not Remedial Discussion **3.** Simplicity as a Vehicle for Generality | - |
| `dMTW` | `W2` | Significance of Performance Gains | **1.** Contextualizing the Gains **2.** Consistency Across a Comprehensive Evaluation **3.** Qualitative Evidence | Section 5.3, Section A.6 |
| `dMTW` | `W3` | Generalizaition Ability | **1.** Successful Generalization to Multi-Codebook VQ (new experiments) **2.** Successful Generalization to Different Model Architectures (new experiments) **3.** Generalization to Other Modalities (future work) | Section A.3, Section A.4 |
| `wwUD` | `W1` | More Comprehensive TTS Results | **1.** New Experiment Results  **2.** Key Findings & Explanation: Smaller Codebooks Benefit TTS; Self-Guidance Enhances Performance | Section 5.4 |
| `wwUD` | `W2,Q` | Explaining the Mechanism | **1.** Clarified Definitions: Quantization Error v.s. Quantization Artifacts v.s. Decoder Robustness **2.** New Visualization as Suggested | Section 3.2, Figure 3 |
| `acqe` | `W1` | Loss Weight Sensitivity Analysis | **1.** Evaluation Across Extended Weight Range **2.** Conclusion: Stable Optimal Range | Section A.2 |
| `acqe` | `W2` | Generalization on Multi-Codebook | **1.** Successful Generalization to Residual FSQ | Section A.3 |
| `acqe` | `W3` | Relationship to Self-Distillation | **1.** Acknowledged Similarity **2.** Clarified Differences: Novel Application & Insight; Architectural & Practical Distinction | Section 2.3 |
| `ckQU` | `W1,Q3` | Significance of Performance Gains | **1.** Comparative Context **2.** Qualitative Evidence **3.** Consistency Across a Comprehensive Evaluation **4.** Focusing on the Core Contribution | Section 5.3 |
| `ckQU` | `W2,Q2` | Generalization on Different Codec Model |  **1.** Successful Generalization to BigCodec | Section A.4|
| `ckQU` | `W3` | Slight WER Increase | **1.** No Consistent Trend **2.** Potential Metric Variance **3.** Consistent Gain in Spectral Intelligibility | Section 5.3 |
| `ckQU` | `Q1` | Feature-Mapping Loss Formulation | **1.** Clarified Formulation Components | Section 4.1 |
| `ckQU` | `Q4` | Influence of Codebook Size on Downstream LLMs | **1.** Text vs. Audio Vocabulary from a Structural Perspective (Core Argument) **2.** Empirical Justification | Section 1 |

---

### Meta-Review · Area_Chair_oFu6 · 2026-01-08

**Summary:**

This paper proposes a new feature-matching loss in the decoder for audio codecs. In the initial reviews, reviewers raised concerns about limited novelty, insignificant performance improvements, and insufficient experimental validation. During the rebuttal, the authors added additional experiments that addressed the concerns regarding experimental coverage; however, the concerns about the novelty of the method and the magnitude of performance gains remain unresolved.

Overall, the AC’s recommendation for this paper is Reject.

**Reviewer Concerns:**

Reviewer dMTW raised concerns about the limited methodological contribution, the lack of convincing empirical improvements across most metrics (including STOI, MCD, WER, SIM, and UTMOS), and the experiments were limited to the audio modality. These concerns were not sufficiently addressed in the rebuttal. The rebuttal did not clearly demonstrate substantial novelty or strong empirical gains over baselines. While the authors discussed possible generalization to other modalities such as images or video, no additional experiments were provided.

Reviewer wwUD expressed concerns regarding the limited novelty of the work and missing experimental validation. The lack of experiments was addressed during the rebuttal through additional results. However, although the authors provided an explanation of the novelty, the significance and impact of the contribution remain unclear, and the concern about limited novelty is only partially resolved.

Reviewer acqe raised concerns about the lack of sensitivity analysis on loss weights, missing experiments involving RVQ, and the absence of discussion of related self-distillation methods. These issues were adequately addressed during the rebuttal.

Reviewer ckQU was concerned about limited performance improvements, missing experiments, and the lack of analysis on WER. The rebuttal showed that performance improvements are consistent but modest and not clearly significant. The missing experiments were addressed by adding results with BigCodec, and the WER analysis was also provided, resolving those concerns.

**Reviewer Scores:**

Reviewer ckQU is likely to increase the score from 4 to 6, while the other reviewers are expected to keep their scores unchanged.

---

### Decision · Program_Chairs · 2026-01-26

Reject